# User Perceptions of Visual Blood: An International Mixed Methods Study on Novel Blood Gas Analysis Visualization

**DOI:** 10.3390/diagnostics13193103

**Published:** 2023-09-30

**Authors:** Greta Gasciauskaite, Justyna Lunkiewicz, Giovanna Schweiger, Alexandra D. Budowski, David Henckert, Tadzio R. Roche, Lisa Bergauer, Patrick Meybohm, Sebastian Hottenrott, Kai Zacharowski, Florian Jürgen Raimann, Eva Rivas, Manuel López-Baamonde, Michael Thomas Ganter, Tanja Schmidt, Christoph B. Nöthiger, David W. Tscholl, Samira Akbas

**Affiliations:** 1Institute of Anesthesiology, University and University Hospital Zurich, Raemistrasse 100, 8091 Zurich, Switzerland; 2Department of Anaesthesiology, Intensive Care, Emergency and Pain Medicine, University Hospital Wuerzburg, University of Wuerzburg, 97080 Wuerzburg, Germany; 3Department of Anaesthesiology, Intensive Care Medicine and Pain Therapy, University Hospital Frankfurt, Goethe University Frankfurt, 60323 Frankfurt, Germany; 4Department of Anaesthesiology, Intensive Care Medicine and Pain Therapy, Hospital Clinic of Barcelona, University of Barcelona, 08036 Barcelona, Spain; 5Institute of Anaesthesiology and Critical Care Medicine, Clinic Hirslanden Zurich, 8032 Zurich, Switzerland

**Keywords:** blood gas analysis, point-of-care diagnostic, qualitative research, situation awareness, user-centered design, visualization, Visual Blood

## Abstract

Blood gas analysis plays a central role in modern medicine. Advances in technology have expanded the range of available parameters and increased the complexity of their interpretation. By applying user-centered design principles, it is possible to reduce the cognitive load associated with interpreting blood gas analysis. In this international, multicenter study, we explored anesthesiologists’ perspectives on Visual Blood, a novel visualization technique for presenting blood gas analysis results. We conducted interviews with participants following two computer-based simulation studies, the first utilizing virtual reality (VR) (50 participants) and the second without VR (70 participants). Employing the template approach, we identified key themes in the interview responses and formulated six statements, which were rated using Likert scales from 1 (strongly disagree) to 5 (strongly agree) in an online questionnaire. The most frequently mentioned theme was the positive usability features of Visual Blood. The online survey revealed that participants found Visual Blood to be an intuitive method for interpreting blood gas analysis (median 4, interquartile range (IQR) 4-4, *p* < 0.001). Participants noted that minimal training was required to effectively learn how to interpret Visual Blood (median 4, IQR 4-4, *p* < 0.001). However, adjustments are necessary to reduce visual overload (median 4, IQR 2-4, *p* < 0.001). Overall, Visual Blood received a favorable response. The strengths and weaknesses derived from these data will help optimize future versions of Visual Blood to improve the presentation of blood gas analysis results.

## 1. Introduction

Medicine continues to grow more and more complex [1]. A continuous gain in life expectancy over the past century, with multiple chronic conditions, combined with new advances in medical science, has increased the difficulty of diagnostic and therapeutic decision making [2,3,4]. This complexity, while necessary for high-quality patient care, plays a critical role in medical error, resulting in increased patient morbidity and mortality [5,6,7,8]. 

Blood gas analysis is illustrative of this change. The three-function blood gas apparatus was introduced into clinical practice in the 1960s, measuring just pH and the partial pressures of carbon dioxide and oxygen in a blood sample [9]. A modern blood gas analyzer, by contrast, measures and calculates many more parameters of gas exchange and acid–base status, as well as hemoglobin, electrolytes, lactate, and glucose. As such, it is useful in diagnosing and managing a wide variety of cardiorespiratory conditions and metabolic derangements [10]. However, despite its widespread use and it being considered a core skill in medicine [11], the interpretation of blood gases is subject to significant error, with the potential for patient harm [12,13].

Better result presentation can help solve this problem. Historically, design has been an afterthought in medicine [14], despite healthcare providers and researchers identifying poor design as a hindrance in daily practice [15]. However, efforts to develop better displays using user-centered design principles have been shown to improve the efficiency and quality of clinical care in various contexts [14,16,17]. Graphical displays are of particular interest in this regard. According to the picture superiority effect, information presented as pictures is better remembered than information presented as words [18]. In clinical settings, this has already been demonstrated using graphical representations of cardiovascular [19] and pulmonary [20] parameters, where they resulted in a faster diagnosis of critical events and more accurate therapy with lower subjective workload. Likewise, our group has demonstrated that graphical and animated representations of vital parameters (Philips Visual Patient Avatar) [21] and rotational thromboelastometry (Visual Clot) are associated with more information transferred per unit of time and lowered perceived workload [22,23].

With this in mind, we developed Visual Blood, a three-dimensional computer animation representing a blood gas analysis printout. We tested Visual Blood for the first time in computer-based simulation studies in 2021–2022—once with a visual reality device and once without it [24,25]. We conducted this mixed qualitative–quantitative study simultaneously to capture the participants’ perceptions and impressions, which help guide further development.

## 2. Materials and Methods

### 2.1. Approval and Consent

Before conducting the study, the responsible local ethics committees in Zurich, Frankfurt, Wuerzburg, and Barcelona reviewed the study protocols and issued declarations of no objection. In addition, we obtained written informed consent from all participants to use the collected data for research purposes. Participation was voluntary, and there was no financial compensation. This article is reported following the SRQR and COREQ guidelines for reporting qualitative research [26,27].

### 2.2. Visual Blood

Visual Blood [24,25] is a novel, three-dimensional representation of blood gas analysis results based on user-centered design principles [28]. The technology presents the animated components of a blood gas analysis represented as icons, including their interactions with each other (Figure 1). Eighteen of the most important parameters of a conventional blood gas analysis are visualized with the technology. Appendix A contains a video which provides a detailed demonstration of how Visual Blood works and how it represents individual parameters within and outside their normal ranges.

### 2.3. Previous Visual Blood Simulation Studies

User perceptions of Visual Blood were collected after two computer-based simulation studies [24,25]. The first one [25] was an international, multicenter, investigator-initiated, prospective, randomized, computer-based simulation study. The study was conducted in five tertiary care hospitals (University Hospital Zurich and Hirslanden Clinic of Zurich in Switzerland, University Hospitals Frankfurt and Wuerzburg in Germany, and Hospital Clinic de Barcelona in Spain) between June and August 2021. After an introduction to Visual Blood and viewing a short instructional video, scenarios of arterial blood gas analysis were shown to care providers as Visual Blood via a virtual reality headset or as a matching conventional printout. The animations were created for the Oculus Quest 2.0 virtual reality headset (Oculus, Meta Platforms, Inc., Menlo Park, CA, USA). After donning an Oculus Quest 2, users find themselves standing in an animation of a blood vessel. The study showed a non-inferiority of Visual Blood compared to the conventional printouts in detecting the deviation of the individual parameters. However, the odds of identifying the correct clinical diagnosis were twice as high with Visual Blood. 

The second study [24] was an international, multicenter, investigator-initiated, prospective, randomized, computer-based simulation study. The study was conducted in three study centers (University Hospital Zurich in Switzerland and University Hospitals Wuerzburg and Frankfurt in Germany) between April and May 2022. After watching an educational video, the participants were presented with Visual Blood animations on a computer screen (without a virtual reality device) or with conventional arterial blood gas printouts. Care providers, even with minimal prior training, demonstrated a higher ability to correctly interpret blood gas analysis results when using Visual Blood. Additionally, their perceived diagnostic confidence was significantly higher compared to working with conventional printouts.

### 2.4. Study Design

We conducted an international, multicenter, researcher-initiated mixed qualitative–quantitative study investigating users’ perceptions of Visual Blood. As participants, we recruited the same anesthesia and intensive care professionals, including staff physicians and residents, who had previously participated in the Visual Blood simulation studies and were now familiar with the new visualization technology. Immediately after the simulations, we interviewed the participants on their positive and negative perceptions of Visual Blood. 

As the second step, the same participants were invited to participate in an online survey, where they rated statements we generated from identified major themes in interview responses on Likert scales.

#### 2.4.1. Qualitative Part: Participant Interviews

Due to the different conditions of the simulation studies—with or without a virtual reality device—the users’ perceptions collected after the two computer-based simulation studies were analyzed separately. This allowed us to examine the impact of various forms of Visual Blood information presentation—with a virtual reality headset and without.

After the simulations, participants were asked to respond to two open-ended questions: ‘What do you LIKE about Visual Blood?’ and ‘What do you DISLIKE about Visual Blood?’ Participants had unlimited time to answer these questions. Following data collection, we initiated our qualitative analysis by translating statements from German and Spanish into English using the online translator deepl.com (DeepL GmbH, Cologne, Germany), accessed on 2 June 2022. In Appendix A, we provide all translated statements from the participants. 

To analyze the insights, we employed thematic analysis and the template approach [27]. Initially, we categorized the statements into positive and negative groups. Then, we grouped individual comments with similar content, resulting in three themes for both the positive and negative statement groups. Figure 2 illustrates the coding template.

Using this coding template, two of the study authors, SA and TRR, conducted categorization after the simulation study with a virtual reality headset. Three of the study authors, GG, GS, and JL, conducted categorization after the simulation study without a virtual reality component. Categorized statements that initially differed were discussed and recategorized through consensus between the reviewers. As recommended in qualitative research, we calculated inter-rater reliability [29,30]. 

#### 2.4.2. Quantitative Part: Online Survey

We conducted an online survey using SurveyMonkey (SVMK Inc., San Mateo, CA, USA) to complement the qualitative results with quantitative data. In this survey, participants were asked to rate six statements (Figure 3) on 5-point Likert scales, ranging from ‘strongly agree’ to ‘strongly disagree.’ These statements were formulated based on the most common themes that emerged from the qualitative phase of this study.

### 2.5. Statistical Analysis

Qualitative data analysis, including coding and data organization, was conducted using Microsoft Word and Excel (Microsoft Corp., Redmond, WA, USA). The results of the qualitative data analysis are presented using descriptive statistics as the number of statements per theme with their respective percentages. We provide inter-rater reliability as a percentage.

The online survey results are presented using descriptive statistics: numbers, medians, and interquartile ranges. Furthermore, we performed a one-sample Wilcoxon signed ranked test (SPSS Inc., Chicago, IL, USA) for each question to determine whether the median value deviates significantly from a neutral point on the Likert scale. Statistical significance was indicated as *p* < 0.05.

## 3. Results

### 3.1. Participant Characteristics

Table 1 provides detailed information about the participants’ characteristics.

### 3.2. Qualitative Analysis

The inter-rater reliability of the coding after the simulation study with a virtual reality headset was over 80%. The inter-rater reliability of the coding after the simulation study without a virtual reality device was 95%.

#### 3.2.1. User Perceptions Following Virtual Reality Simulations

After the simulation study, where participants interpreted blood gas analysis in a virtual reality environment [25], the information overview, intuitiveness, innovativeness, and timesaving aspects of Visual Blood were emphasized as the most appreciated features of the technology. Nevertheless, using a virtual reality device made interpretation challenging—animations moved too quickly, creating an unpleasant sensation in three-dimensional space and, thus, leading to cognitive overload. Table 2 presents a summary of the exact number of perceived advantages and disadvantages by theme. 

The theme addressed most often was the positive usability features (40%, or 68 out of 165 statements). For example, participant 22 noted that Visual Blood is “easy to memorize with the help of visualizations”, and participant 30 specified “the electrolyte and acid–base status is quick and easy to recognize”. The potential for improvement in usability (27%, or 44 out of 165 statements) was mainly described as an opportunity to reduce the cognitive burden the clinicians have to deal with when interpreting complex blood gas analysis results further. For instance, participant 3 stated that “if several imbalances are present, a bit of concentration is needed,” leading to “information overload”.

The design features were also well received. Overall, the advantages of the technology’s design were highlighted in 10% (17 out of 165) of statements. Participant 25, for example, appreciated “color coding of the individual values, clear representation of each value at a glance”, while participant 47 liked the “emphasis on the values to focus on”. Some other participants highlighted aspects where the design could still be improved (11%, or 18 out of 165 statements). For instance, participant 10 found the “hemoglobin variables a bit confusing”, while participant 20 found the “colors difficult to differentiate”.

Concerning the study design, only 2% of statements (3 out of 165) were favorable, whereas 9% (15 out of 165 statements) criticized it, emphasizing that the virtual reality device made the interpretation of results challenging. Participants who liked the study design commented that “the VR experience was a nice change” (participant 27) and “good explanatory video, easy handling” (participant 29). The negative comments mainly identified the training period or study period as “too short” (participant 29) to get used to Visual Blood, primarily due to the challenges posed by the virtual reality device.

#### 3.2.2. User Perceptions Following Simulations (No Virtual Reality Headset)

As in the simulation study with a virtual reality component, the theme that received the most comments here was also the positive usability features (31%, or 62 out of 200), especially emphasizing the technology’s intuitiveness and timesaving qualities. For instance, participant 1 stated that Visual Blood is “easy to use”, whereas participant 65 pointed out that “with a bit of practice, pathologies can be detected more quickly than with conventional blood gas analysis”. Comments indicating the potential for usability improvement (19.5%, or 39 out of 200) were mainly focused on the amount of information presented simultaneously on the screen and missing data quantification. For example, participant 19 stated that there is “very much information on a small space”, whereas participant 23 pointed out that he must “still look at a conventional blood gas analysis afterwards” because “numbers are missing”.

The design of Visual Blood was also well appreciated (18.5%, or 37 out of 200 statements). Participant 42 highlighted positive design features, emphasizing the “creative and colorful presentation” of the information provided by Visual Blood, whereas participant 63 agreed that the technology offers a “good graphical information presentation”. Several ideas for improving the technology have also been identified (24%, or 48 out of 200); for example, participant 41 pointed out that “rapid flashing has a stressful effect”, and participant 34 said that the “flow of molecules is too fast”.

Concerning the study design, two themes—time pressure and training video—were emphasized the most by participants. For instance, participant 59 said that “the interpretation time of 15s is a bit short”, whereas participant 2 pointed out that “Definitions of pathological versus normal states after watching the introduction video only once cannot be clearly remembered for all parameters”.

Table 2 summarizes the exact number of perceived advantages and disadvantages by theme. Figure 4 provides a percentage distribution of all statements by theme collected after the simulation studies with and without a virtual reality headset.

### 3.3. Online Survey (Quantitative Part)

After completing the qualitative analysis, we invited the same participants to participate in the online survey. Figure 3 illustrates the formulation of the questions in the online survey and presents the results.

## 4. Discussion

This international, multicenter, researcher-initiated mixed qualitative–quantitative study explored participants’ impressions of a novel, three-dimensional blood gas analysis display—Visual Blood. We analyzed user feedback after two computer-based simulation studies—the first with a virtual reality device and the second without it. User perceptions are essential when identifying the positive aspects of the technology and considering the potential for improvement in the future. 

The principal qualitative findings demonstrate that the most common theme we derived from the field notes after both simulation studies was usability. There were more positive than negative comments concerning this topic. This finding was then confirmed in our online survey as part of our quantitative analysis, where participants agreed that Visual Blood was intuitive and could be used with minimal training. This is in line with previous work, which has shown that graphical displays are more efficient at transferring information and often preferred by end users [22]. It also validates the user-centered design approach taken in developing Visual Blood.

The Visual Blood design received favorable feedback. However, participants in both the interviews and the online survey provided valuable insights that will be beneficial for the future development of the technology. For instance, while many appreciated the color-coding of the parameters, some participants found it challenging to differentiate between colors. One explanation could be a potential case of color blindness. However, we cannot provide detailed information about this, as it was not within the scope of our inquiry. The color design in Visual Blood will be crucial for further development.

By contrast, the study design, particularly the use of the virtual reality device in the first simulation study described above, received significant criticism. Many participants had little to no prior experience with this relatively new technology, leading to side effects known as simulator sickness, including discomfort and nausea, in some participants [31]. This highlights the importance of ongoing development and refinement of virtual reality devices, which are increasingly being integrated into the medical field, particularly in intensive care [32]. These criticisms and observations provide invaluable insights for the ongoing improvement of Visual Blood.

Notably, the results demonstrate a strong initial level of acceptance of Visual Blood in its first version, especially when compared to traditional conventional blood gas analysis printouts. Nevertheless, there is considerable room for further improvement. Given the foundation of good usability and positive feedback, it is reasonable to assume that both the design and study design can be further optimized, thereby enhancing the overall user experience of Visual Blood.

This study has several strengths and limitations. There are typical limitations of qualitative research in the interview part of the study. The results of the qualitative analysis cannot be generalized to larger populations as confidently as quantitative results due to the absence of statistical significance testing [33]. However, the online survey, the quantitative part of the study, provided further insight into the main themes identified. 

Moreover, this study was conducted in a controlled, computer-based environment free from stress and distractions. During the study, we presented the blood gas analysis printout and Visual Blood for precisely 15 s. These conditions differ from real-world scenarios in clinical practice, where physicians face stress, distractions, and can decide how long to examine a blood gas analysis printout.

Additionally, Visual Blood relies on a three-dimensional representation that requires a virtual reality headset. While it could theoretically be introduced to the operating room (OR), the current practicality of working in the OR with a VR headset is limited. Strategies such as strategically placing VR glasses in the OR may be explored to address this issue.

We consider the high data quality in this study a strength, as field notes were collected immediately after participants completed the Visual Blood simulation studies, ensuring accurate and meaningful qualitative data. Furthermore, the multicenter and international design enhances the external validity of our results and allows for extrapolation to a wide range of international healthcare professionals. The randomized order and presentation of multiple scenarios with the two different methods enabled the most accurate comparison possible.

## 5. Conclusions

We designed this study to gather qualitative information to inform the further development of Visual Blood, a new user-centered design-based technology for visualizing blood gas analysis results. Within the context of our study, Visual Blood received broad positive feedback. A detailed analysis of participants’ statements revealed that they considered Visual Blood an innovative and intuitive visualization technique for blood gas analysis results. However, our results also highlight the need for improvement, particularly in optimizing Visual Blood’s design to reduce visual overload.

In a world and medical field increasingly characterized by complexity, it becomes essential to find ways to integrate technology into everyday medical practice to assist clinicians. Emerging user-centered, design-based technologies, such as Visual Blood, hold promise in this regard. This study marks an important milestone in the development of Visual Blood, demonstrating its potential as a clinical aid. However, further development and research are required before it can be integrated into daily clinical practice.

## Figures and Tables

**Figure 1 diagnostics-13-03103-f001:**
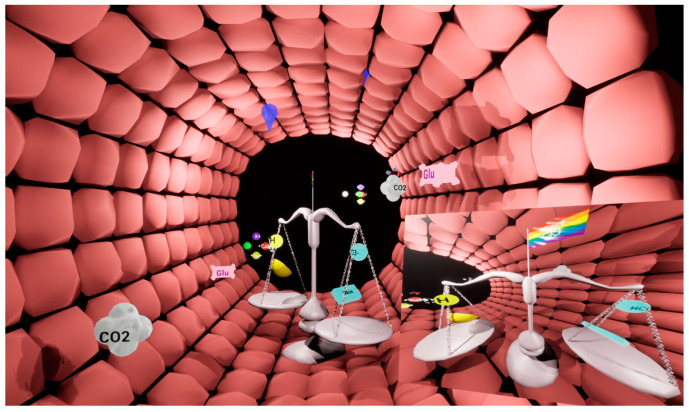
This image of Visual Blood depicts a blood vessel’s interior with various parameters in their normal ranges. In the background, you can see a droplet representing serum osmolarity, a glucose molecule (in pink), a carbon dioxide molecule (in grey), and electrolyte concentrations. The scale with an acid/proton on one side and a base/bicarbonate ion on the other signifies the acid–base balance.

**Figure 2 diagnostics-13-03103-f002:**
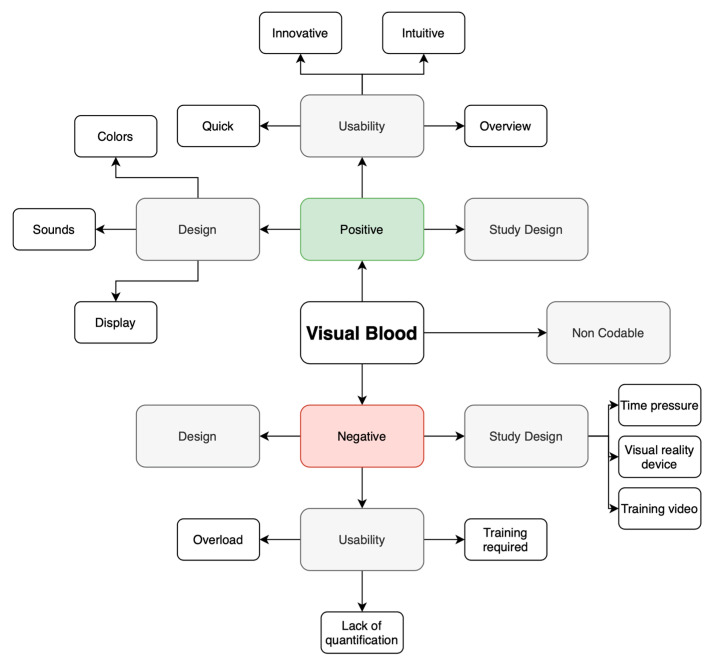
A coding template representing themes for positive and negative user perceptions.

**Figure 3 diagnostics-13-03103-f003:**
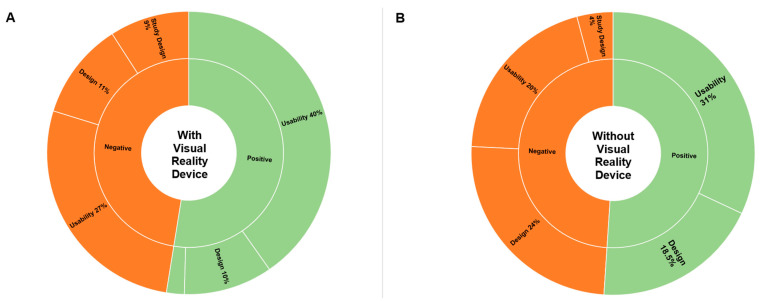
Sunburst diagrams of the coding tree, presenting a percentage distribution of all statements by theme. The length of each category corresponds to its percentage representation in the statements. The inner and outer circles represent two coding levels. The innermost circle corresponds to the highest level of categorization, while the outermost circle corresponds to subcategorization in different themes. (**A**) Sunburst diagram, representing a distribution of themes in simulations with a visual reality device. (**B**) Sunburst diagram, representing a distribution of themes in simulations without a visual reality device.

**Figure 4 diagnostics-13-03103-f004:**
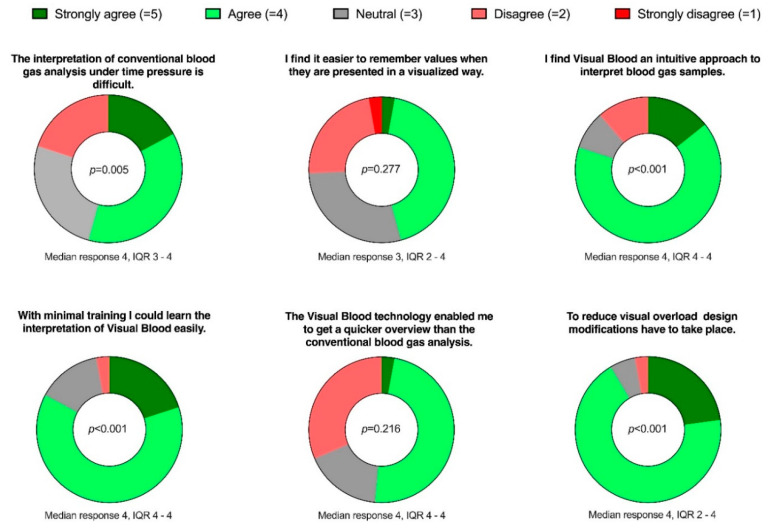
Donut charts representing responses to the six online survey questions. The *p*-value, calculated using a one-sample Wilcoxon signed-rank test, indicates deviations from the ‘neutral’ answer.

**Table 1 diagnostics-13-03103-t001:** Study and participant characteristics.

Study and Participant Characteristics	Values
Simulations with visual reality device	
Study centers, n	5
Participants per study center, n	10
Female participants, n (%)	31 (62%)
Age (years), median (*IQR)	31 (28–41)
Senior physicians, n (%)	22 (44%)
Resident physicians, n (%)	28 (56%)
Total work experience (years), median (IQR)	5 (2–10)
Blood gas analysis skills, self-rated, (0 = novice, 100 = expert), median (IQR)	70.5 (60–83)
Video game playing, subjective frequency (0 = never, 100 = very often), median (IQR)	6 (0–31)
Simulations without visual reality device	
Study centers, n Participants at University Hospital Zurich in Switzerland Participants at University Hospital Wuerzburg Participants at University Hospital Frankfurt in Germany	3 35 18 17
Female participants, n (%)	42 (60%)
Age (years), median (IQR)	30 (28–39)
Senior physicians, n (%)	15 (21.4%)
Resident physicians, n (%)	55 (78.6%)
Total work experience (years), median (IQR)	6 (3–8)
Blood gas analysis skills, self-rated, (0 = novice, 100 = expert), median (IQR)	29 (22–37)

*IQR—interquartile range.

**Table 2 diagnostics-13-03103-t002:** Major themes, participant count, percentage, and examples.

Users’ Perceptions after the Computer-Based Simulations with Virtual Reality	Users’ Perceptions after the Computer-Based Simulations without Virtual Reality
Positive (87/165, 53%) Design (17/165, 10%) ○ Color coding of the individual values, clear representation of each value at a glance (*P 25). ○ Auditory input (P 5). ○ Clear, visualized physiology of simple numerical values in the body (blood vessel) from my patient (P 39). Usability (68/165, 40%) ○ Main diagnosis easier to see [than with conventional blood gas analysis] (P 23). ○ Easier to see the problem in a short time [than with conventional blood gas analysis] (P 43). ○ More intuitive approach [than conventional blood gas analysis] (P 6). Study design (3/165, 2%) ○ The VR experience was a nice change (P 27).	Positive (103/200, 52%) Design (37/200, 19%) ○ Colorful (P 13). ○ Lactate funny, illustrative presented (P 15). ○ Visually appealing (P 16). Usability (62/200, 31%) ○ Fast overview over pathological results (P 42). ○ Simultaneous seeing and perception of pathologically altered blood gas analyses (P 47). ○ Time saving (P 16).
Negative (78/165 47%) Design (18/165, 11%) ○ Colors difficult to differentiate (P 20). Usability (44/165, 27%) ○ If several imbalances are present, a bit of concentration is needed (P 3). ○ No absolute values visible, extent of pathologies more difficult to quantify (P 28). ○ Would need training and practice to be useful in everyday clinical practice (P 2). Study design (15/165, 9%) ○ Too little time [to get used to Visual Blood] (P 16).	Negative (97/200 48.5%) Design (48/200, 24%) ○ Slow speed of the individual components swimming through (P 17). ○ Too much motion (P 20). ○ pH value representation could be clearer (P 36). Usability (39/200, 20%) ○ Too overloaded and confusing when many parameters are out of the normal range (P 65). ○ Absolute values necessary in addition (P 2). ○ Need to get used to it (P 22). Study design (8/200, 4%) ○ With many pathological parameters, an overview is difficult in 15 s (P 50). ○ Time pressure (P 59). ○ Definitions of pathological versus normal states after watching the introduction video only once not clearly remembered for all parameters (P 47).

*P: Participant number.

## Data Availability

The complete datasets are available upon reasonable request.

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
