# Peer review of "User Perceptions of Visual Blood: An International Mixed Methods Study on Novel Blood Gas Analysis Visualization"

_diagnostics, 2023, doi:10.3390/diagnostics13193103_

Round 1

Reviewer 1 Report

Provided animation to discuss how visual blood work, is interesting.

Some comments:

1. All acronyms should be defined at first appearance in the abstract and manuscript: as IQR in line 30.

2. Please improve quality of Figs.1, 2 and 3.

3. On line 299, numerous participants have highlighted the challenge posed by differences in colors. I believe developers could consider addressing this issue by exploring ways to enhance their techniques for individuals who may experience color blindness.

4. You should think about how to interpretate qualitative results to quantitative in a better and simple way. This is important.

5. When it comes to virtual reality (VR), it's essential to consider how operators can perform tasks for extended periods without encountering any issues. Overall, it should be user-friendly.
